# Ionizing Radiation Protein Biomarkers in Normal Tissue and Their Correlation to Radiosensitivity: Protocol for a Systematic Review

**DOI:** 10.3390/jpm11010003

**Published:** 2020-12-22

**Authors:** Anne Dietz, Maria Gomolka, Simone Moertl, Prabal Subedi

**Affiliations:** Bundesamt für Strahlenschutz/Federal Office for Radiation Protection, Ingolstädter Landstraße 1, 85764 Oberschleissheim, Germany; adietz@bfs.de (A.D.); mgomolka@bfs.de (M.G.); smoertl@bfs.de (S.M.)

**Keywords:** ionizing radiation, normal tissue, biomarker, radiotherapy, radiosensitivity, protein

## Abstract

*Background:* Radiosensitivity is a significantly enhanced reaction of cells, tissues, organs or organisms to ionizing radiation (IR). During radiotherapy, surrounding normal tissue radiosensitivity often limits the radiation dose that can be applied to the tumour, resulting in suboptimal tumour control or adverse effects on the life quality of survivors. Predicting radiosensitivity is a component of personalized medicine, which will help medical professionals allocate radiation therapy decisions for effective tumour treatment. So far, there are no reviews of the current literature that explore the relationship between proteomic changes after IR exposure and normal tissue radiosensitivity systematically. *Objectives:* The main objective of this protocol is to specify the search and evaluation strategy for a forthcoming systematic review (SR) dealing with the effects of in vivo and in vitro IR exposure on the proteome of human normal tissue with focus on radiosensitivity. *Methods:* The SR framework has been developed following the guidelines established in the National Toxicology Program/Office of Health Assessment and Translation (NTP/OHAT) Handbook for Conducting a Literature-Based Health Assessment, which provides a standardised methodology to implement the Grading of Recommendations Assessment, Development and Evaluation (GRADE) approach to environmental health assessments. The protocol will be registered in PROSPERO, an open source protocol registration system, to guarantee transparency. *Eligibility criteria:* Only experimental studies, in vivo and in vitro, investigating effects of ionizing radiation on the proteome of human normal tissue correlated with radio sensitivity will be included. Eligible studies will include English peer reviewed articles with publication dates from 2011–2020 which are sources of primary data. *Information sources:* The search strings will be applied to the scientific literature databases PubMed and Web of Science. The reference lists of included studies will also be manually searched. *Data extraction and results:* Data will be extracted according to a pre-defined modality and compiled in a narrative report following guidelines presented as a “Synthesis without Meta-analyses” method. *Risk of bias:* The risk of bias will be assessed based on the NTP/OHAT risk of bias rating tool for human and animal studies (OHAT 2019). *Level of evidence rating*: A comprehensive assessment of the quality of evidence for both in vivo and in vitro studies will be followed, by assigning a confidence rating to the literature. This is followed by translation into a rating on the level of evidence (high, moderate, low, or inadequate) regarding the research question. Registration: PROSPERO Submission ID 220064.

## 1. Introduction

### 1.1. Background and Rationale

The International Agency for Research on Cancer (IARC) Global Cancer Observatory reports more than 18 million new cases of cancer in 2018 [1] and radiotherapy (RT) is used to treat 50–60% of cancers [2]. The delivered dose during standard RT is balanced between optimal tumor kill and avoidance of damage to surrounding tissues [3]. Depending on the tumor entity, up to 20% of patients show a moderate to severe detrimental response to ionizing radiation (IR) treatment [4]. Acute effects include erythema, inflammation and mucositis depending on cancer type while late effects are typically fibrosis, atrophy, vascular damage and neurocognitive and endocrine dysfunctions, especially for brain irradiated children [5,6,7]. These side effects limit radiation doses that can be applied to the tumor, often leading to suboptimal tumor control or to serious impairment of the quality of life of survivors. In a small subset of patients the severe reactions can be ascribed to known radiation hypersensitivity syndromes, such as Ataxia–Telangiectasia (A-T), Fanconi anemia (FA), or Nijmegen Breakage Syndrome (NBS) [8,9,10]. As late as 2010, children with AT mutations have succumbed to death following RT [11]. These genetic syndromes, however, only comprise about 1% of the patients demonstrating severe side effects [12]. Therefore, most of the normal tissue reactions cannot be explained by known genetic disorders and no clear guidelines exist for medical doctors to individually adapt their therapy scheme. This has led to an increased interest in predicting personalized radiosensitivity.

Radiosensitivity is any enhanced tissue or cell reaction after a subject has been exposed to IR when compared to the majority of other “normal” responding individuals [13,14]. The reactions include inflammation, fibrosis, cardiovascular illness, cataracts, and cognitive decline [15]. Individual radiosensitivity can be applied as component of personalized medicine in RT. Personalized medicine is not about finding out novel medications but sub-dividing individuals in various subgroups that vary in their response to treatment for a specific disease [16]. It is performed to tailor the treatment to the individual need. This implies that medical professionals can target cancer patients, who are radiosensitive, with lower doses and alternative treatment schedules, for example, chemotherapy rather than radiotherapy. On the other hand, patients who are less radiosensitive could be given higher doses of IR to maximize the likelihood of treatment success [14]. Moreover, although the potential for therapy using ionizing radiation is unparalleled, there is an increasing concern for the risks posed by low-dose occupational exposure among workers in nuclear industries and healthcare [17,18,19].

It was already established in the early twentieth century that individuals respond differently to IR [20] and the reason behind this is still under thorough investigation. Discovering breakthroughs in individualized radiosensitivity is difficult because the effect of IR is modified by age, gender, lifestyle, genetic predisposition, and the quality and quantity of IR dose these individuals receive [21]. Several conventional reviews that summarize IR-induced changes at a molecular level have been published over the years. For example, a compilation of cytogenetic damage, epigenomic alterations, induced and germline mutations, DNA and nucleotide pool damage, and transcriptomic and translational biomarkers of radiation exposure for epidemiological studies have been reported [22]. Similarly, DNA double stand break repairs, chromosomal aberrations and radiation-induced apoptosis in ex vivo irradiated blood lymphocytes as predictors of radiosensitivity have also been described [23]. A review of various proteomics approaches to investigate cancer radiotherapy in cancer and normal cell lines and in bio fluids of in vivo irradiated individuals has also been performed [24].

This review is different to conventional reviews—it is a systematic review (SR). Unlike conventional reviews, SR provides an unbiased selection of studies that include an objective and transparent evaluation of the evidence. [25,26]. SR begins with defining the terms PECO—population, exposure, comparators, and outcome, which helps to produce a well formulated research question for the SR. Each of the terms has an inclusion and exclusion criteria, which furthermore specifies which studies will be included. Each study is then evaluated for the relationship between exposure and outcome, as well as dissected based on questions that define selection, confounding, performance, attrition or exclusion, detection, and selective reporting risk of biases [27,28].

Out of 28,279 studies (PubMed search term: ionizing radiation [Title/Abstract], retrieval date 23 September 2020) there are no systematic reviews (SRs) that investigate proteomic changes after exposure to IR. Taking into consideration that the study of Pernot et al. [22] including protein biomarkers for IR exposure took place in 2012, we have compiled proteomic markers of radiosensitivity in normal tissues from the last 10 years (2011–2020). In this article we provide a protocol that determines our search and evaluation strategy for the actual systematic review.

Our planned review aims at presenting the status quo of IR-induced changes in protein expression in normal tissue that can be correlated to radiosensitivity, which can be used to further investigate the concept of individual radiosensitivity. This will help to personalize treatment strategies for cancer patients during radiotherapy (RT) or help to assist an individualized risk assessment process by identifying and protecting occupationally exposed persons l nuclear workers and radiologists. A future issue may be the protection of sensitive cosmonauts from harmful effects of cosmic radiation.

### 1.2. Objectives

The main objective of this SR is to evaluate the effects of ionizing radiation on the proteome of human normal tissue regarding radiosensitivity in experimental models (in vivo and in vitro).

Following sub-objectives will be taken into account:Narrative presentation of the current status of knowledge obtained from experimental studies (in vivo, in vitro) evaluating the effect of ionizing radiation on the proteome of human normal tissue regarding radiosensitivity.Evaluation of the quality of evidence using a confidence rating and establishing a level of evidence for the presence or absence of a biomarker for radio sensitivity in human normal tissue.

## 2. Methods

This structure for this systematic review, as presented in the graphical abstract, was adapted according to the National Toxicology Program/Office of Health Assessment and Translation NTP/OHAT handbook [27], which provides standard operating procedures for conducting a systematic review and integrating evidence [26,29]. To rate the quality of the scientific evidence, GRADE (Grading of Recommendations Assessment, Development and Evaluation) will be used. This is a formal process that is often used in systematic reviews, which is also applied to develop recommendations in guidelines that are as evidence-based as possible [25,29]. The selection of articles, data extraction and synthesis, as well as risk of bias assessment, will be performed manually. The synthesis of data will be performed narratively without meta-analysis, as explained by Campbell et al. [30]. This SR will adhere strictly to PRISMA (Preferred Reporting Items for Systematic Reviews and Meta-Analyses) guidelines [28,31], which provide an evidence-based minimum set of items that need to be reported for evaluation of randomized trials or can be used as a basis to judge other research types, e.g., evaluations of interventions. The protocol and the abstract will also be reported as described in PRISMA-P [32] and PRISMA-A [33] respectively.

The SR is registered in the International Prospective Register of Systematic Reviews on 10 November 2020 (submission ID 220064)

### 2.1. Eligibility Criteria

Studies that comply with elements of PECO (Population, Exposure, Comparators, and Outcome as outlined in Table 1) will be included in this SR.

#### 2.1.1. Population

The population in this SR will include both in vivo and in vitro models.

In vivo models: This model will include humans or blood, biopsies, and body fluids taken from humans.

In vitro models: This model will include non-cancer tissue culture, primary human non-cancer cell lines, or non-cancer cell lines derived from humans.

This review will exclude non-human studies. This review will also exclude tumour cell lines, tumour tissue, and biopsies. The rationale behind excluding tumour data is that we focus on radiation induced effects in normal tissue. Although some mechanisms and pathways may overlap, there are also clear differences of radiation resistance and sensitivity mechanisms in tumour compared to normal tissue. The protein markers identified here should help to predict radiation sensitivity reactions of normal tissue of cancer patients and therefore assist a personalized radiation therapy treatment. In addition, identified markers can help in risk assessment of radiation exposed individuals, such as nuclear workers, accidentally exposed individuals, and individuals living in areas of higher background ionizing radiation.

#### 2.1.2. Exposure

The exposure will be ionizing radiation (IR). The World Health Organization defines IR as radiation with enough energy that during an interaction with an atom, it can remove tightly bound electrons from atoms, which results in the atom being charged or ionized [34]. Therefore, this study will include all sources of IR: X-Ray, cosmic rays, gamma ray, alpha and beta particles, carbon and proton therapy, and all sources of natural background ionizing radiation. This review will exclude non-ionizing radiation (infrared, near-infrared, ultraviolet, microwaves, electromagnetic radiation or radio waves)

#### 2.1.3. Comparators

Comparators in this study will be humans or in vitro models that have not been exposed to IR.

In case of studies including humans exposed to IR, material such as blood, before and after IR, taken from the same human, will be included. When tissue samples before and after irradiation are compared the localisation of the samples within the radiation field must be ensured and dose estimates should be provided.

In case of studies involving humans living in areas of higher-than-average natural background radiation, comparators will be humans that live in areas of average natural background radiation but from a similar demographic community.

Studies that do not have a comparator group will be excluded.

#### 2.1.4. Outcomes

Outcomes of interest are changes in protein expression levels correlated to radiosensitivity. We have explained before that radiosensitivity could include inflammation, fibrosis, cardiovascular illness, and cognitive decline. Therefore, studies that have included information on such parameters will be included.

In Vivo models should report overall survival and in vitro models should mention survival, apoptosis, proliferation, colony formation or metabolic assays.

#### 2.1.5. Exclusion criteria prioritisation

Studies will be excluded if they are:Not a primary studyIrrelevant population: in vivo: not human data; in vitro: not human-derived cell lines, tumour cell lines, tumour tissues (or cell lines derived from tumour tissues)Irrelevant Exposure (not ionizing radiation)Irrelevant Outcome (not studies related to protein expression that correlates to radiosensitivity)Studies not in English, or no full text for the study is availableStudies outside time frame (not within 2011–2020)

### 2.2. Search Strategy

#### 2.2.1. Databases

The searches will be performed in NCBI PubMed [35] (https://pubmed.ncbi.nlm.nih.gov/) and ISI Web of Knowledge v.5.34 [36] (https://www.webofknowledge.com/). Any additional study might be added manually later. The references will be imported into Microsoft Excel and the duplicates removed. The search string for ISI Web of Knowledge is provided in Appendix A with this protocol.

#### 2.2.2. Search Strings

The search strings will be a combination of population, exposure, and outcome elements from the PECO parameters. The population of interest are human and/or normal tissues, the exposure of interest is ionizing radiation, and the outcome of interest is ‘radiosensitivity and the corresponding changes in protein expression’. Restriction for language and time period will be set where studies published between 2011 and 2020 in English will be considered. Search strings that were used in Web of Science are presented in Appendix A and the strings will be adapted and calibrated to be used in PubMed.

#### 2.2.3. Study Selection

Studies will be subjected to a two-phase screening, which is also presented in the graphical abstract. As a Phase I screening, AD and PS will together cross-check the title, abstract, and the key words with the inclusion/exclusion criteria. Retained articles will be downloaded for a phase II full-text screening manually. Any article excluded in Phase II screening, along with the reason for exclusion, will be recorded and provided in the Appendix A. Any disagreements between the reviewers will be solved by consensus, involving MG or SM if necessary.

### 2.3. Data Extraction

Data extraction will be performed by PS and AD together and any discrepancies will be solved by consensus. Google sheets will be used to enter the data and the result will be finally reported in Microsoft Excel. The form for data extraction is provided in Appendix A.

### 2.4. Body of Evidence Structure

Evidence will be organised in outcome-related groups favouring data synthesis (proteins) and confidence rating at the health outcome level (radiation induced normal tissue radio sensitivity).

The criteria to determine the inclusion of specific studies or experiments in each outcome group will consider the evidence stream (i.e., in vivo, in vitro), health outcomes/endpoints or exposure regime (high or low dose, duration). Two outcomes, primary and secondary, have been defined. According to the literature, IR induces toxicity in normal tissue and in some cases shows a radio sensitive phenotype. This is the definition of primary outcome in this review. Secondary outcomes represent intermediary endpoints upstream of primary outcomes. Altered protein expression after exposure to IR, which may be grouped around specific signalling or functional pathways, has been defined as the secondary outcome.

### 2.5. Internal Quality Assessment

The included studies will be internally quality controlled using the Risk of Bias (RoB) tool developed by the Office of Health Assessment and Translation [27,37]. The RoB tool acts as an internal quality control in reviewing the articles included for the SR. Even though the studies follow a methodological flow, they might still have a bias, which might lead to an underestimation or an overestimation of the effect of the exposure. For example, if the population contains mainly older subjects and the comparator contains younger ones, the effect of the exposure might be overestimated. To critically evaluate the studies, the following questions will be asked:

#### 2.5.1. Selection Bias

Was administered dose or exposure level adequately randomised?Was allocation to study groups adequately concealed?Did selection of study participants result in appropriate comparison groups?

#### 2.5.2. Confounding Bias

4.Did the study design or analyses account for important confounding and modifying variables?

#### 2.5.3. Performance Bias

5.Were experimental conditions identical across study groups?6.Were the research personnel and human subjects blinded to the study group in the study?

#### 2.5.4. Attrition/Exclusion Bias

7.Were outcome data complete without attrition or exclusion from analysis?

#### 2.5.5. Detection Bias

8.Can we be confident in the exposure characterization?9.Can we be confident in the outcome assessment?

#### 2.5.6. Selective Reporting Bias

10.Were all measured outcomes reported?

#### 2.5.7. Other Sources of Bias

11.Were there any other potential threats to internal validity (e.g., statistical methods were appropriate and researchers adhered to study protocol?)

Each question will be answered with ‘definitely low risk of bias’, ‘probably low risk of bias’, ‘probably high risk of bias’, or ‘definitely high risk of bias’. Responses will be determined together by PS and AD. Considering the RoB responses of each question, the study will be categorized into three tiers, T1, T2 and T3, as proposed by the National Toxicology Program/Office for Health Assessment and Translation (NTP/OHAT). The tiers will be based on ‘Key Questions’, which are domains of randomization bias, outcome detection bias, and performance bias.

### 2.6. Confidence Rating for Each Body of Evidence

Extracted data from each study included will be considered as independent bodies of evidence. An assessment will be performed for each to define a confidence rating. The confidence rating reflects the reliability with which the study findings accurately depict a true effect of IR toxicity on normal tissue and linkage to radiosensitivity, as described in the NTP/OHAT handbook [27]. Each body of evidence is given an initial confidence rating that is downgraded or upgraded according to factors that decrease or increase confidence in the results.

Initial confidence rating is based on the presence or absence of four features. The features are (1) controlled exposure (2) exposure prior to outcome (3) individual outcome data and (4) use of comparison group

The ratings are as follow:High—4 features (++++)Moderate—3 features (+++)Low—2 features (++)Very Low—1 feature (+)

The initial confidence is then upgraded or downgraded depending on certain factors, and a confidence in the body of evidence is provided (Table 2). The factors increasing confidence are magnitude, dose response, and consistency across studies and the factors decreasing confidence are risk of bias, unexplained inconsistency, and imprecision.

### 2.7. Translation of Confidence Rating Into Level of Evidence (for the Health Effect)

Ionizing radiation leads to toxicity in all living organisms, in both tumour and non-tumour cells. Therefore, the translation of confidence rating into levels of evidence will not be performed in the review. This SR aims to investigate proteomic changes in normal tissues and our population consists of in vitro studies of primary human material and immortalized cell lines as well as in vivo studies, on occupationally, naturally and accidentally exposed persons and on radiotherapy patients. The defined population in this SR are exposed to a broad range of radiation qualities, IR doses and dose-rates, and focusing on one particular type of dose and dose would hamper the objective of the study. Therefore, no preference for high or low-doses, respectively, or high or low-dose rates will be made. The different doses and dose-rates, however, will be provided for all selected studies in the final review.

### 2.8. Data Synthesis

Included data will likely comprise randomized and non-randomized trials. Moreover, clinical diversity is inevitable because of the defined PECO parameters as effects of IR on expression of different proteins being investigated. The secondary outcome is an altered protein expression after a population has been exposed to IR. It is highly probable that studies look at a diverse set of proteins, and studies will not have proteins in common that show an altered expression. Therefore, a meta-analyses might not be possible and in that case synthesis of results is performed in a narrative way or described textually. No reporting guidelines exist for narrative synthesis and although it provides a clear picture of the effects of exposure, such synthesis lacks transparency [38]. When methods other than meta-analyses are used to synthesize results, certain findings, such as reporting of synthesis structure and comparison grouping, standardised metric used for synthesis, synthesis method, presentation of data, and the summary of the synthesis finding, are left unreported. To counter these problems, data synthesis will be reported using the narrative synthesis without meta-analyses (SWiM) method, as presented by Campbell et al. [30].

### 2.9. Differences between Protocol and the Review

If there are any methodological deviations from this protocol in the review to be written, they will be mentioned in the ‘Differences between protocol and review’.

### 2.10. Potential Applications of the Protocol/Review

The application or outcome of this protocol is the definition of parameters for the systematic evaluation of protein changes which are correlated with radiosensitivity in normal tissue. The primary outcome of the systematic review is the identification of these proteins. In the review article we will also discuss potential applications of these proteins in medical practice.

## Figures and Tables

**Table 1 jpm-11-00003-t001:** Population, exposure, comparators, and outcome (PECO) Statement with inclusion and exclusion criteria.

PECO		Inclusion Criteria	Exclusion Criteria
Population	In Vivo	Humans	Non-human
	In Vitro	Human tissues, primary human non-tumour cell line, derived human non-tumour cell line	Tumour cells and tissues, primary and secondary tumour cell lines
Exposure	In Vivo	Ionizing radiation (e.g., alpha and beta particles, X-Rays, Gamma rays, proton therapy)	Non-ionizing radiation (e.g., radio and microwaves, near infrared, ultraviolet, electromagnetic waves)
	In vitro		
Comparators	In Vivo	Non-exposed humans, bio fluids before IR	Lacks control group
	In Vitro	Non-exposed cells or tissues	
Outcomes	In Vivo	Changes in protein expression correlated with radiosensitivity	Irrelevant outcome (e.g., changes in transcriptome, or protein changes not correlated to radiosensitivity)
	In vitro		

**Table 2 jpm-11-00003-t002:** Accessing confidence in body of evidence.

Initial Confidence by Key Features of Study Design	Factors Decreasing Confidence	Factors Increasing Confidence	Confidence in the Body of Evidence
High (++++) 4 features	Risk of bias	Magnitude effect	High (++++)
Moderate (+++) 3 features	Unexplained inconsistency	Dose response	Moderate (+++)
Low (++) 2 features	Indirectness	Residual confounding	Low (++)
Very low (+) ≤1 feature	Imprecision	Consistency	Very low (+)

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
