# Peer review of "Ionizing Radiation Protein Biomarkers in Normal Tissue and Their Correlation to Radiosensitivity: Protocol for a Systematic Review"

_jpm, 2020, doi:10.3390/jpm11010003_

Round 1
Reviewer 1 Report
In this protocol, The authors have aimed to design a robust protocol to evaluate the effect of radiation on the proteome of human normal tissue in vivo and in vitro based on existing literature. While the premise is intriguing and the design of this protocol is promising, the manuscript suffers from shortcomings that need to be addressed.
The two major issues:
- The terms protocol and reviews have been used across the manuscript and it gives the impression that this is a review which is not the case. It is a protocol paper based on which a review will be written in the future. This needs to be clarified.
- The authors want to include proton radiation under the ionizing radiation category. While this is technically correct, proton radiation is unique in a way that it is designed to delivered in a targeted fashion and reduce damage to the surrounding healthy tissue. A great of its use is in pediatric CNS tumors, where proton theory has been shown to reduce long-term morbid side effects such as cognitive deficiency, endocrine issues, etc. There, I am sure how much sense does testing the effect of proton radiation on healthy tissue/cells make since healthy areas are not the intended targets for proton radiation.
- Please see my detailed comments file attached.
Author Response
1. The authors understand that differentiating between the protocol and the systematic review (SR) to be written could be confusing. Therefore, we have added this information also in the manuscript.
Abstract, Page 1, Lines 19-21
“Objectives: The main objective of this protocol is to specify the search and evaluation strategy for a forthcoming systematic review (SR) dealing with the effects of in vivo and in vitro IR exposure on the proteome of human normal tissue with focus on radiosensitivity.”
Introduction, Page 3, Lines 95-96
In this article we provide a protocol that determines our search and evaluation strategy for the actual systematic review.
2. Disclaimer: The attached file was not available, also after the editor was asked.
The authors agree that proton therapy is an attractive method to reduce toxicities of radiotherapy because of the decrease of integral radiation dose to normal tissues. Nevertheless we think proton irradiation effects on normal tissues are important for several reasons. Firstly, range uncertainties and necessary clinical safety margins together with scattered neutrons may result in radiation effects in the surrounding normal tissue. Secondly, proton radiation induced systemic effects by irradiated circulating blood cells and tumor secreted factors should be considered. Beside clinical applications, the understanding of proton radiation effects on normal tissue is an important issue for radiation protection during space missions. In the review, the different sources of ionizing radiation used in the study will be provided.
Reviewer 2 Report
The manuscript “Ionizing radiation protein biomarkers in normal tissue and their correlation to radiosensitivity: protocol for a systematic review” provides a very good basis for writing a systemic review. The authors provide a transparent and understandable description of the methodology, linking it to a highly relevant topic in the field of radiation biology.
I very much support the publication of this great written review in a present form.
Author Response
The authors would like to thank the reviewer for the encouraging words.
Reviewer 3 Report
The manuscript presents the protocol for a systematic review aiming to correlate various protein biomarkers, with the radiosensitivity. Overall, the protocol is well presented, with sufficient details on the inclusion and exclusion criteria etc.
Further details and refinements to the proposed protocol may provide a more comprehensive review, are detailed below:
- Firstly, radiosensitivity usually refers to the intrinsic response of cells and tissues to radiation. Various models are used in radiation biology to assign numerical parameters to account for specific radiosensitivity. These aspects are not discussed here and appear not to be accounted for, especially when discussing the in vitro studies.
- The aim of this study may benefit from further clarification, as to provide a clear rationale for the omission of any tumour data in the study
- The authors include a very broad spectrum of irradiation, and the protocol does not seem to correct for radiation quality and delivery method (targeted irradiation, whole body irradiation in the case of cosmic radiation). These aspects are known to induce different effects on the cells/organs.
- Why have the studies using human tumour-derived cells have been excluded? Most of modern radiation studies will focus on a potential interaction between normal and tumour tissues, pf great importance when discussing targeted irradiation
- The authors do not differentiate between high and low dose rates within the literature cases. Why is that? Considering there is significant evidence pointing towards the dose rate as a significant modulator for the radiation effects.
- Is the analysis accounting for any potential drug-radiation interactions?
Author Response
1. As a part of our search algorithm, we have included the term ‘sensitv*’ in our search parameters (Supporting information: search algorithm). Any studies that have the term ‘sensitiv*’ in their abstract/title are automatically included in our review, provided the study also confirms to our other PECO (population, exposure, comparators, and outcome) terms. Therefore, only studies that are correlated to clinical radiation sensitivity or intrinsic radiation sensitivity as determined by viability tests, clonogenic survival, apoptotic assays are included.
They are mentioned in the lines 180-181 in the manuscript.
2. The authors thank the reviewer for this suggestion. The rationale behind excluding tumour data is that we focus on radiation induced effects in normal tissue. Although some mechanisms and pathways may overlap there are also clear differences of radiation resistance and sensitivity mechanisms in tumour compared to normal tissue. However systemic effects influenced by the tumour and tumour tissue interaction will be accounted for by including tumour secreted factors in our study.
We have included this in the manuscript.
2.1. Eligibility criteria, Page 4, (Lines 150-157)
“The rationale behind excluding tumour data is that we focus on radiation induced effects in normal tissue. Although some mechanisms and pathways may overlap there are also clear differences of radiation resistance and sensitivity mechanisms in tumour compared to normal tissue. The protein markers identified here should help to predict radiation sensitivity reactions of normal tissue of cancer patients and therefore assist a personalized radiation therapy treatment. In addition identified marker can help in risk assessment of radiation exposed individuals, as nuclear workers, accidentally exposed individuals, and individuals living in areas of higher background ionizing radiation.”
3. We agree with the reviewer that the quality and quantity of irradiation produce different reactions in cells, tissues, and organs. However, in this SR we intend to identify radiation-induced protein changes related to radiosensitivity in normal tissue without further limitations. Therefore, we decided to include a broad spectrum of irradiation. Provided that we identify an appropriate number of publications, we will specify the protein changes to different radiation conditions in the SR.
4. We agree that the potential interaction between normal and tumour tissues is of great importance while investigating normal tissue radiosensitivity in regards to exposure after IR and definitely requires further investigation. However, the pathways that lead to sensitivity after IR in tumour and normal tissues are not necessarily similar.
While tumour tissues have been excluded, studies that have investigated proteomic changes in body fluids (blood including PMBCs and extracellular vesicles, plasma, and urine) of cancer patients who have undergone radiotherapy, are included. (see also question #2)
5. We included high and low dose rates in our initial search to get a broad overview about all radiation induced protein changes in normal tissue. Focusing on specific dose rates would hamper the envisaged inclusion of a broad study population.
If we will find enough suitable studies, we will subdivide and prioritize the results into high-dose rate proteomic changes and low-dose rate proteomic changes.
We have included the answer to this question also in the manuscript.
2.6 Translation of confidence rating into level of evidence, Page 8, Lines 297-303
“This SR aims to investigate proteomic changes in normal tissues and our population consists of in vitro studies of primary human material and immortalized cell lines as well as in vivo studies, on occupationally, naturally and accidentally exposed persons and on radiotherapy patients. The defined population in this SR are exposed to a broad range of radiation qualities, IR doses and dose-rates and focusing on one particular type of dose and dose would hamper the objective of the study. Therefore, no preference for high or low-doses respectively high or low-dose rates will be done. The different doses and dose-rates, however, will be provided for all selected studies in the final review.”
6. The reviewer refers to an important yet overlooked part in many studies: drug-radiation interactions. In our study, this part falls under the risk of bias (RoB) questions of if there are confounding bias (question no 4: Did the study design or analyses account for important confounding and modifying variables). We do not perform any further analyses per se, but we rate a study with a higher risk of bias if they have not performed analyses for in vivo subjects if they were a part of chemotherapy or if the in vitro samples were accompanied with anti-cancer drugs.
Reviewer 4 Report
Dose, dose rate, and fractionation are significant factors in ionizing radiation outcomes. It would be beneficial to include these criteria in the review. If they are excluded, a rationale should be provided.
Author Response
Please compare answer to question (#3) from Reviewer 3
A rationale has been provided in the manuscript.
2.6 Translation of confidence rating into level of evidence, Page 8, Lines 297-303
“This SR aims to investigate proteomic changes in normal tissues and our population consists of in vitro studies of primary human material and immortalized cell lines as well as in vivo studies, on occupationally, naturally and accidentally exposed persons and on radiotherapy patients. The defined population in this SR are exposed to a broad range of radiation qualities, IR doses and dose-rates and focusing on one particular type of dose and dose would hamper the objective of the study. Therefore, no preference for high or low-doses respectively high or low-dose rates will be done. The different doses and dose-rates, however, will be provided for all selected studies in the final review.”
Therefore, no selective preference will be performed between high and low-doses or high and low-dose rates. The different dose and dose-rates, however, will be provided for all selected studies in the final review.
Round 2
Reviewer 1 Report
Dear authors,
I am sorry to hear that you never received my attached review report. I really enjoyed reading your manuscript. Please find my comments below and as an attachment.
Radiotherapy (RT) is used to treat about 50-60 % of cancers [1], comprising up to about 8 million radiotherapy patients per year worldwide (UNSCEAR 2008) [2].
-Is there an updated citation for ref 2? It is about a decade old.
Objectives: The main objective of this systematic review (SR) is to evaluate the effect of in vivo and in 19 vitro exposure to IR on the proteome of human normal tissue regarding radiosensitivity.
-The objective of this manuscript is a bit unclear. In the beginning, it gives the impression that this is supposed to be a systematic review. However, this is a protocol for a systematic review. The authors mention at the end of the manuscript under section 2.8 that the review is still to be written. The authors need to clarify this upfront to avoid confusion for the readers.
For example, children with AT (ataxia–telangiectasia) mutations have succumbed to death as late as 2010 following RT [4]. This happens because in subjects with A-T mutation, there is a lack of functional ATM gene, which results in an inability to activate damage-inducible DNA repair, thereby triggering programmed cell death following non lethal doses of IR [5].
-This is indeed a great example. However, the authors should include other examples of IR such as IR induced long-term morbid side effects in children with CNS brain tumors. Plus it would be great to include examples of some adult cancers.
Although some of these extreme radiation sensitivity syndromes are known
-This is a little bit of a vague sentence here, since it is not clear what is the authors referring to “some of these extreme radiation sensitivity syndromes”. The authors have mentioned ‘ death’ as one of the syndromes and despite its severity, it is still one of the side effects of IR. Adding more examples of IR sensitivity syndromes will help justify using the phrase “some of these”.
It was already established in early twentieth century that individuals respond differently to IR [10] and the reason behind is still under thorough investigations
-The authors probably imply ‘21st century’ since reference 10 was published in 2016.
One of the reasons is because IR 67 affects individuals differently on multiple scales including age, gender, lifestyle, diet, genetic 68 predisposition, and quality and quantity of IR dose received [11].
-This sentence does not make much sense (probably because of how it is constructed)
Unfortunately, none of aforementioned reviews were conducted systematically .
-The authors need to elaborate a bit about what they imply by “ lack of systematic review”. They have explained SR in the next paragraph, however, it seems to be a bit disconnected and not cohesive. A reshuffling of the sentences would be ideal.
The authors should precisely outline the ‘unmet need’ and how this protocol and the following review will help address that unmet need.
Methods
The systematic review will be performed as described in the Cochrane Handbook[16], explained in further detail by Rooney et al. [19], and will be reported as per Preferred Reporting Items for 102 Systematic Reviews and Meta-Analyses (PRISMA) guidelines[18,20]
-Despite the citations, the authors need to explain the methods a bit further even though briefly. Since the backbone of this systematic review is based on established protocols cited by the authors, it makes sense to briefly explain the review method/protocol.
The results of phase II screening for the eligibility of articles, extraction of data and the risk of bias tool will be reported in Microsoft 107 Excel and submitted along with the review for transparency.
-The phase II kind of comes out of nowhere without any mention about the phase I screening
Exposure
- Are proton therapies included? The authors should include a comprehensive list of types of ionization radiations included in the study instead of giving examples. Proton therapies could be used as an ideal example of ionizing radiation that deposits maximal energy into the targeted area, minimally affecting surrounding healthy tissue.
Ok, as I read through, I see that proton therapy is indeed included in the study. This should be made clear in the table. Otherwise, the purpose of the table is not solved. Also, the authors need to consider, how much of a sense does including proton beam data in in-vitro studies make since proton beams tend to deposit maximum energy in the target (usually in tumors) and then rapidly lose energy as they enter a healthy region. This phenomenon is hard to replicate in vitro.
Comparators
Comparators in this study will be humans or in vitro models that have not been exposed to IR.
In the case of studies including humans that will be exposed to IR, material such as blood, before and after IR, taken from the same human, will be included [6].
-The purpose of this reference is not clear
Tissues taken before and after radiation 155 should be from the immediate vicinity
-Please define ‘immediate vicinity’ for eg with a certain mm of distance?
The authors should include a section about potential applications of this protocol/ the review to be written in the field of personalized medicine and how the findings can be translated into the clinic.
Author Response
Dear reviewer,
Thank you for the insightful comments and suggestions, which we have incorporated in our protocol. Below, we provide a point-by-point answer to the questions.
- Radiotherapy (RT) is used to treat about 50-60 % of cancers [1], comprising up to about 8 million radiotherapy patients per year worldwide (UNSCEAR 2008) [2].
-Is there an updated citation for ref 2? It is about a decade old.
A newer reference has been added and the sentence has been rephrased (, Page 2, Lines 48-49)
“IARC Global Cancer Observatory reports more than 18 million new cases of cancer in 2018 (https://gco.iarc.fr/) and radiotherapy (RT) is used to treat 50-60% of cancers [1]”
- Objectives: The main objective of this systematic review (SR) is to evaluate the effect of in vivo and in 19 vitro exposure to IR on the proteome of human normal tissue regarding radiosensitivity.
-The objective of this manuscript is a bit unclear. In the beginning, it gives the impression that this is supposed to be a systematic review. However, this is a protocol for a systematic review. The authors mention at the end of the manuscript under section 2.8 that the review is still to be written. The authors need to clarify this upfront to avoid confusion for the readers.
Please also see our answer for question #1 in the previous version of the answer.
The authors understand that differentiating between the protocol and the systematic review (SR) to be written could be confusing. Therefore, we have added this information also in the manuscript.
(Page 1, Lines 19-21)
“Objectives: The main objective of this protocol is to specify the search and evaluation strategy for a forthcoming systematic review (SR) dealing with the effects of in vivo and in vitro IR exposure on the proteome of human normal tissue with focus on radiosensitivity.”
Page 3, Lines 121-122)
“In this article we provide a protocol that determines our search and evaluation strategy for the actual systematic review.”
- For example, children with AT (ataxia–telangiectasia) mutations have succumbed to death as late as 2010 following RT [4]. This happens because in subjects with A-T mutation, there is a lack of functional ATM gene, which results in an inability to activate damage-inducible DNA repair, thereby triggering programmed cell death following non lethal doses of IR [5].
-This is indeed a great example. However, the authors should include other examples of IR such as IR induced long-term morbid side effects in children with CNS brain tumors. Plus it would be great to include examples of some adult cancers.
We have combined the answer to question #3 and question #4 by adding the following in introduction. (Page 2, Line53-66)
“Depending on the tumour entity, up to 20 % of the patients show a moderate to severe detrimental response to ionizing radiation (IR) treatment [2]. Acute effects include erythema, inflammation and mucositis depending on cancer type while late effects are typically fibrosis, atrophy, vascular damage and neurocognitive and endocrine dysfunctions, especially for brain irradiated children [3-5]. These side effects limit radiation doses that can be applied to the tumour, often leading to suboptimal tumour control or to serious impairment of the quality of life of survivors. In a small subset of patients the severe reactions can be ascribed to known radiation hypersensitivity syndromes, such as Ataxia–Telangiectasia (A-T), Fanconi anemia (FA), or Nijmegen Breakage Syndrome (NBS) [6-8]. As late as 2010, children with AT mutations have succumbed to death following RT [9]. These genetic syndromes however only comprise about 1% of the patients demonstrating severe side effects [10]. Therefore most of the normal tissue reactions cannot be explained by known genetic disorders and no clear guidelines exist for medical doctors to individually adapt their therapy scheme. This has led to an increased interest in predicting personalized radiosensitivity.”
- Although some of these extreme radiation sensitivity syndromes are known
-This is a little bit of a vague sentence here, since it is not clear what is the authors referring to “some of these extreme radiation sensitivity syndromes”. The authors have mentioned ‘ death’ as one of the syndromes and despite its severity, it is still one of the side effects of IR. Adding more examples of IR sensitivity syndromes will help justify using the phrase “some of these”.
Please refer to our answer for question #3
- It was already established in early twentieth century that individuals respond differently to IR [10] and the reason behind is still under thorough investigations
-The authors probably imply ‘21st century’ since reference 10 was published in 2016.
We apologize for the wrong reference. The correct reference is
Bouchacourt, M. L., Sur la difference de sensibilite aux rayons de Roentgen de la peau des differents sujets, et, sur le meme sujet des differents regions du corps. Sciences 1911, 942-947.
The reference has also been updated in the manuscript.
- One of the reasons is because IR 67 affects individuals differently on multiple scales including age, gender, lifestyle, diet, genetic 68 predisposition, and quality and quantity of IR dose received [11].
-This sentence does not make much sense (probably because of how it is constructed)
This sentence has been rewritten as follow in the manuscript. (Page 2-3, Lines 87-89)
“Discovering breakthroughs in individualized radiosensitivity is difficult because the effect of IR is modified by age, gender, lifestyle, genetic predisposition, and the quality and quantity of IR dose these individuals receive. “
- Unfortunately, none of aforementioned reviews were conducted systematically .
-The authors need to elaborate a bit about what they imply by “lack of systematic review”. They have explained SR in the next paragraph, however, it seems to be a bit disconnected and not cohesive. A reshuffling of the sentences would be ideal.
The sentences have been reshuffled (Page 3-4, Lines 91-122)
Many conventional reviews to summarize IR-induced changes on a molecular level have been published over the years. For example, a compilation of cytogenetic damage, epigenomic alterations, induced and germline mutations, DNA and nucleotide pool damage, and transcriptomic and translational biomarkers of radiation exposure for epidemiological studies have been reported [11]. Similarly, DNA double stand break repairs, chromosomal aberrations and radiation-induced apoptosis in ex vivo irradiated blood lymphocytes as predictors of radiosensitivity have also been described [12]. A review of various proteomics approaches to investigate cancer radiotherapy in cancer and normal cell lines and in bio fluids of in vivo irradiated individuals has been performed as well [13].
This review is different to conventional reviews- it is a systematic review (SR). Unlike conventional reviews, SR provides an unbiased selection of studies that include an objective and transparent evaluation of the evidence. [14,15]. SR begins with defining the terms PECO- population, exposure, comparators, and outcome, which helps well formulated research question of the SR. Each of the term has an inclusion and exclusion criteria, which furthermore specifies which studies will be included. Each study is then evaluated for the relationship between exposure and outcome as well as dissected based on questions that define selection, confounding, performance, attrition or exclusion, detection, and selective reporting risk of biases [16,17].
- The authors should precisely outline the ‘unmet need’ and how this protocol and the following review will help address that unmet need.
Unmet need has been added in the manuscript (Page 3, Line 124-130)
“Our planned review aims at presenting the status quo of IR-induced changes in protein expression in normal tissue that can be correlated to radiosensitivity, which can be used to further investigate the concept of individual radiosensitivity. This will help to personalize treatment strategies to cancer patients during radiotherapy (RT) or help to assist an individualized risk assessment process by identifying and protecting occupationally exposed persons like nuclear workers and radiologists. A future issue may be the protection of sensitive cosmonauts from harmful effects of cosmic radiation. “
- The systematic review will be performed as described in the Cochrane Handbook[16], explained in further detail by Rooney et al. [19], and will be reported as per Preferred Reporting Items for 102 Systematic Reviews and Meta-Analyses (PRISMA) guidelines[18,20]
-Despite the citations, the authors need to explain the methods a bit further even though briefly. Since the backbone of this systematic review is based on established protocols cited by the authors, it makes sense to briefly explain the review method/protocol.
We have defined it further in the manuscript (Page 4, Lines 150-162)
“This structure for this systematic review, as presented in graphical abstract, was adapted according to NTP/OHAT handbook [16], which provides standard operating procedures for conducting a systematic review and integrating evidence[15,18]. To rate the quality of the scientific evidence, GRADE (Grading of Recommendations Assessment, Development and Evaluation) will be used. It is a formal process that is often used in systematic reviews, which is also applied to develop recommendations in guidelines that are as evidence- based as possible [14,18]. The selection of articles, data extraction and synthesis, as well as risk of bias assessment will be performed manually. The synthesis of data will be performed narratively without meta-analysis, as explained by Campbell and co-workers [19]. This SR will adhere strictly to PRISMA (Preferred Reporting Items for Systematic Reviews and Meta-Analyses) guidelines [17,20], which provide an evidence-based minimum set of items that needs to be reported for evaluation of randomized trials, or can be used as a basis to judge other research types, e.g. evaluations of interventions. The protocol and the abstract will also be reported as described in PRISMA-P [21] and PRISMA-A [22] respectively.”
- The results of phase II screening for the eligibility of articles, extraction of data and the risk of bias tool will be reported in Microsoft 107 Excel and submitted along with the review for transparency.
-The phase II kind of comes out of nowhere without any mention about the phase I screening
To make the sentence clearer, we have rephrased the paragraph (Page 7, Lines 262-267)
“Studies will be subjected to a two-phase screening, which is also presented in the graphical abstract. As a Phase I screening, AD and PS will together cross-check the title, abstract, and the key words with the inclusion/exclusion criteria. Retained articles will be downloaded for a phase II full-text screening manually. Any article that will be excluded in Phase II screening, along with the reason for exclusion, will be recorded and provided in the supplementary information. Any disagreements between the reviewers will be solved by consensus and involving MG or SM if necessary”
- Are proton therapies included? The authors should include a comprehensive list of types of ionization radiations included in the study instead of giving examples. Proton therapies could be used as an ideal example of ionizing radiation that deposits maximal energy into the targeted area, minimally affecting surrounding healthy tissue.
Ok, as I read through, I see that proton therapy is indeed included in the study. This should be made clear in the table. Otherwise, the purpose of the table is not solved. Also, the authors need to consider, how much of a sense does including proton beam data in in-vitro studies make since proton beams tend to deposit maximum energy in the target (usually in tumors) and then rapidly lose energy as they enter a healthy region. This phenomenon is hard to replicate in vitro.
Please compare our answer to question#2 in the previous version of the answer.
The authors agree that proton therapy is an attractive method to reduce toxicities of radiotherapy because of the decrease of integral radiation dose to normal tissues. Nevertheless we think proton irradiation effects on normal tissues are important for several reasons. Firstly, range uncertainties and necessary clinical safety margins together with scattered neutrons may result in radiation effects in the surrounding normal tissue. Secondly, proton radiation induced systemic effects by irradiated circulating blood cells and tumor secreted factors should be considered. Beside clinical applications, the understanding of proton radiation effects on normal tissue is an important issue for radiation protection during space missions. In the review, the different sources of ionizing radiation used in the study will be provided.
The term proton therapy has been also included in Table 1. (Page 5)
Comparators
12, Comparators in this study will be humans or in vitro models that have not been exposed to IR.
In the case of studies including humans that will be exposed to IR, material such as blood, before and after IR, taken from the same human, will be included [6].
-The purpose of this reference is not clear
Thank you for overseeing this. It was a typographical error and the reference has been removed.
- Tissues taken before and after radiation 155 should be from the immediate vicinity
-Please define ‘immediate vicinity’ for eg with a certain mm of distance?
Immediate vicinity for tissue samples before/after irradiation increase the validity of the data. In our experiments on skin biopsies we are inside a maximum distance of 1 cm to the initial cut. Nevertheless we see the practical difficulties of this demand, especially for tissues other than skin. Therefore we changed this point and added the following sentence (Page 6, Lines 217-219)
“When tissue samples before and after irradiation are compared the localisation of the samples within the radiation field must be ensured and dose estimates should be provided.”
- The authors should include a section about potential applications of this protocol/ the review to be written in the field of personalized medicine and how the findings can be translated into the clinic.
A section about potential application has been added to the manuscript (Line 380-384, Page 10)
“The application or outcome of this protocol is the definition of parameters for the systematic evaluation of protein changes which are correlated with radiosensitivity in normal tissue. The primary outcome of the systematic review is then the identification of these proteins. In the review article we will also discuss potential applications of these proteins in medical practice”
References mentioned in this document
- Rosenblatt, E.; Izewska, J.; Anacak, Y.; Pynda, Y.; Scalliet, P.; Boniol, M.; Autier, P. Radiotherapy capacity in European countries: an analysis of the Directory of Radiotherapy Centres (DIRAC) database. The Lancet Oncology 2013, 14, e79-e86, doi:10.1016/s1470-2045(12)70556-9.
- Seibold, P.; Auvinen, A.; Averbeck, D.; Bourguignon, M.; Hartikainen, J.M.; Hoeschen, C.; Laurent, O.; Noël, G.; Sabatier, L.; Salomaa, S., et al. Clinical and epidemiological observations on individual radiation sensitivity and susceptibility. International Journal of Radiation Biology 2019, 96, 324-339, doi:10.1080/09553002.2019.1665209.
- Bledsoe, J.C. Effects of Cranial Radiation on Structural and Functional Brain Development in Pediatric Brain Tumors. Journal of Pediatric Neuropsychology 2015, 2, 3-13, doi:10.1007/s40817-015-0008-2.
- Follin, C.; Erfurth, E.M. Long-Term Effect of Cranial Radiotherapy on Pituitary-Hypothalamus Area in Childhood Acute Lymphoblastic Leukemia Survivors. Curr Treat Options Oncol 2016, 17, 50, doi:10.1007/s11864-016-0426-0.
- Barnett, G.C.; West, C.M.; Dunning, A.M.; Elliott, R.M.; Coles, C.E.; Pharoah, P.D.; Burnet, N.G. Normal tissue reactions to radiotherapy: towards tailoring treatment dose by genotype. Nat Rev Cancer 2009, 9, 134-142, doi:10.1038/nrc2587.
- Nakanishi, K.; Taniguchi, T.; Ranganathan, V.; New, H.V.; Moreau, L.A.; Stotsky, M.; Mathew, C.G.; Kastan, M.B.; Weaver, D.T.; D'Andrea, A.D. Interaction of FANCD2 and NBS1 in the DNA damage response. Nat Cell Biol 2002, 4, 913-920, doi:10.1038/ncb879.
- Petrini, J.H. The mammalian Mre11-Rad50-nbs1 protein complex: integration of functions in the cellular DNA-damage response. Am J Hum Genet 1999, 64, 1264-1269, doi:10.1086/302391.
- Digweed, M. Human genetic instability syndromes: single gene defects with increased risk of cancer. Toxicology Letters 1993, 67, 259-281, doi:10.1016/0378-4274(93)90061-2.
- Pietrucha, B.M.; Heropolitanska-Pliszka, E.; Wakulinska, A.; Skopczynska, H.; Gatti, R.A.; Bernatowska, E. Ataxia-telangiectasia with hyper-IgM and Wilms tumor: fatal reaction to irradiation. J Pediatr Hematol Oncol 2010, 32, e28-30, doi:10.1097/MPH.0b013e3181bfd3d9.
- Mizutani, S.; Takagi, M. XCIND as a genetic disease of X-irradiation hypersensitivity and cancer susceptibility. Int J Hematol 2013, 97, 37-42, doi:10.1007/s12185-012-1240-5.
- Pernot, E.; Hall, J.; Baatout, S.; Benotmane, M.A.; Blanchardon, E.; Bouffler, S.; El Saghire, H.; Gomolka, M.; Guertler, A.; Harms-Ringdahl, M., et al. Ionizing radiation biomarkers for potential use in epidemiological studies. Mutat Res 2012, 751, 258-286, doi:10.1016/j.mrrev.2012.05.003.
- Chua, M.L.; Rothkamm, K. Biomarkers of radiation exposure: can they predict normal tissue radiosensitivity? Clin Oncol (R Coll Radiol) 2013, 25, 610-616, doi:10.1016/j.clon.2013.06.010.
- Azimzadeh, O.; Tapio, S. Proteomics approaches to investigate cancer radiotherapy outcome: slow train coming. Translational Cancer Research 2017, 6, S779-S788, doi:10.21037/tcr.2017.03.83.
- Rooney, A.A.; Cooper, G.S.; Jahnke, G.D.; Lam, J.; Morgan, R.L.; Boyles, A.L.; Ratcliffe, J.M.; Kraft, A.D.; Schunemann, H.J.; Schwingl, P., et al. How credible are the study results? Evaluating and applying internal validity tools to literature-based assessments of environmental health hazards. Environ Int 2016, 92-93, 617-629, doi:10.1016/j.envint.2016.01.005.
- Higgins, J.P.T.; Thomas, J.; Chandler, J.; Cumpston, M.; Li, T.; Page, M.J.W., V. A. . Cochrane Handbook for systematic reviews of interventions, 2 ed.; John Wiley & Sons: Chichester (UK), 2019.
- Program, N.T. Handbook for Conducting a Literature-Based Health Assessment Using OHAT Approach for Systematic Review and Evidence Integration. Services, N.T.P.U.S.D.o.H.a.H., Ed. U.S.A., 2019.
- Liberati, A.; Altman, D.G.; Tetzlaff, J.; Mulrow, C.; Gotzsche, P.C.; Ioannidis, J.P.; Clarke, M.; Devereaux, P.J.; Kleijnen, J.; Moher, D. The PRISMA statement for reporting systematic reviews and meta-analyses of studies that evaluate health care interventions: explanation and elaboration. PLoS Med 2009, 6, e1000100, doi:10.1371/journal.pmed.1000100.
- Rooney, A.A.; Boyles, A.L.; Wolfe, M.S.; Bucher, J.R.; Thayer, K.A. Systematic review and evidence integration for literature-based environmental health science assessments. Environ Health Perspect 2014, 122, 711-718, doi:10.1289/ehp.1307972.
- Campbell, M.; McKenzie, J.E.; Sowden, A.; Katikireddi, S.V.; Brennan, S.E.; Ellis, S.; Hartmann-Boyce, J.; Ryan, R.; Shepperd, S.; Thomas, J., et al. Synthesis without meta-analysis (SWiM) in systematic reviews: reporting guideline. BMJ 2020, 368, l6890, doi:10.1136/bmj.l6890.
- Moher, D.; Liberati, A.; Tetzlaff, J.; Altman, D.G.; Group, P. Preferred reporting items for systematic reviews and meta-analyses: the PRISMA statement. PLoS Med 2009, 6, e1000097, doi:10.1371/journal.pmed.1000097.
- Shamseer, L.; Moher, D.; Clarke, M.; Ghersi, D.; Liberati, A.; Petticrew, M.; Shekelle, P.; Stewart, L.A.; Group, P.-P. Preferred reporting items for systematic review and meta-analysis protocols (PRISMA-P) 2015: elaboration and explanation. BMJ 2015, 350, g7647, doi:10.1136/bmj.g7647.
- Beller, E.M.; Glasziou, P.P.; Altman, D.G.; Hopewell, S.; Bastian, H.; Chalmers, I.; Gotzsche, P.C.; Lasserson, T.; Tovey, D.; Group, P.f.A. PRISMA for Abstracts: reporting systematic reviews in journal and conference abstracts. PLoS Med 2013, 10, e1001419, doi:10.1371/journal.pmed.1001419.
Round 3
Reviewer 1 Report
I thank the authors for their diligent rebuttal and revision of the manuscript. It is now ready for acceptance.